# Percolation and Internet Science

**Franco Bagnoli** [1,2,]*[ID], **Emanuele Bellini** [3][ID], **Emanuele Massaro** [4,5][ID] **and Raúl Rechtman** [6][ID]

1 Department of Physics and Astronomy and CSDC, University of Florence, via G. Sansone 1, 50019 Sesto Fiorentino, Italy

2 INFN, sez. Firenze, via G. Sansone 1, 50019 Sesto Fiorentino, Italy

3 Centre on Cyber-Physical Systems (C2PS), Khalifa University, Saada Street, P.O. Box 127788, Abu Dhabi, United Arab Emirates; belliniema@gmail.com

4 HERUS Lab, École Polytechnique Fédérale de Lausanne (EFPL), GR C1 455 (Bâtiment GR)—Station 2, CH-1015 Lausanne, Switzerland; emanuele.massaro@epfl.ch

5 ISI Foundation, via Chisole 5, 10126 Torino, Italy

6 Instituto de Energás Renovables, Universidad Nacional Autónoma de México, Temixco 62580, Mexico; rrs@ier.unam.mx

\* Correspondence: franco.bagnoli@unifi.it

**Abstract:** Percolation, in its most general interpretation, refers to the "flow" of something (a physical agent, data or information) in a network, possibly accompanied by some nonlinear dynamical processes on the network nodes (sometimes denoted reaction–diffusion systems, voter or opinion formation models, etc.). Originated in the domain of theoretical and matter physics, it has many applications in epidemiology, sociology and, of course, computer and Internet sciences. In this review, we illustrate some aspects of percolation theory and its generalization, cellular automata and briefly discuss their relationship with equilibrium systems (Ising and Potts models). We present a model of opinion spreading, the role of the topology of the network to induce coherent oscillations and the influence (and advantages) of risk perception for stopping epidemics. The models and computational tools that are briefly presented here have an application to the filtering of tainted information in automatic trading. Finally, we introduce the open problem of controlling percolation and other processes on distributed systems.

**Keywords:** stochastic processes; networks; risk perception; opinion modeling; epidemic modeling

## 1. Introduction

The Internet is naturally linked to the network concept, i.e., a set of nodes connected by links [1–4]. Clearly, this model is ubiquitous and intensively studied: how a structured network arises and grows, how it can be crossed, cut or made more robust, it can be explored and measured, and how processes are affected by the network structure. However, the Internet should not be seen only at the interconnection level. As pointed out in [5], we can put into evidence at least two aspects related to networks: the physical/infrastructural one, related to protocols and transferring of information, and the social/human one, which uses the infrastructural level to communicate, create and elaborate information (see, for instance, the proceedings of the *Internet Science* conference for some examples of multi-disciplinary approaches [6–9]). According to this, Internet science can be considered a research field that investigates the aspect of the communication among actors and resources as processes that can shape the information relay and transformations on the Internet [5]. This field brings together, among others, contributions from networks and complexity sciences.

One of the crucial contributions of the network science to Internet investigations is the study of the diffusion or "flow" of something on the network itself, which can be broadly associated to the

word "percolation" [10–14]. A simple definition of a percolation problem is the one referring to the porosity of some material, and the possibility for a fluid to pass through it, if there is at least a path connecting two opposite sides. However, one does not need to have a pre-defined network. One can examine the problem of joining sites with links, and ask whether the resulting pattern is composed by many isolated sub-networks or if most of nodes belongs to the same cluster.

The percolation problem is a cornerstone of statistical mechanics and stochastic processes. In this review, we illustrate some of the ideas coming from the investigations about percolation and cellular automata, which can be of interest for Internet science.

A lattice or graph is a set of $N$ nodes with links between some nodes. The nodes can be visualized as pores and the links as pipes on a substrate or material. All channels are open, but pores may be open with probability $p$ and closed with probability $1 - p$. One asks if there exists an open path connecting any two sides of the material. Clearly, for $p = 0$, there is no open path, while, for $p = 1$, all paths are open, and therefore there is a critical value $p_c$ marking the transition between the existence or absence of such a path. All results are considered in the limit $N \to \infty$.

Let us define a computational model for such a problem. We assign to each node $i$ of a lattice a random number $r_i$ uniformly distributed between zero and one. Then, we mark all sites with $r_i < p$. All marked points connected by one link are said to belong to a cluster, and the number $s$ of connected nodes define the size of such cluster. For each value of $p$, we have a certain distribution $P(s)$ of clusters ($P(s)$ is proportional to the number of clusters of size $s$ present on the graph). Since the numbers $r_i$ do not change (they are said to form a *quenched* random field), the clusters can only grow and merge when increasing $p$. For small $p$, we have an exponential distribution $P(s)$ peaked on a small value of $s$. For large values of $p$, we have a large cluster (giant component), spanning over almost all the lattice and including almost all the sites in the graph/lattice, with a tail of small clusters. Excluding the giant component, the distribution of clusters is again exponential.

The state corresponding to the critical values of $p$, $p_c$, is characterized by the appearance of the giant component. Just below it, the average cluster size diverges, and the cluster size distribution becomes much flatter, changing from an exponential to a power law. This is reminiscent of the fluctuations observed in thermal systems (such as the Ising model) near a continuous phase transition.

Indeed, one can introduce correlation functions that behave in a way similar to their thermal equivalent, allowing the evaluation of the critical exponents for such models. In principle, these results depend on the choice of the random numbers $r_i$, and should be averaged over many realizations of the random field. However, in many cases, these systems show a property called "selfa-veraging", i.e., the quantities computed on a large enough system are the same as those averaged over many realizations. This is certainly true far from the critical value $p_c$, since in these cases the correlation functions decrease exponentially with a well-defined correlation length. Patches that are farther away than this correlation length are uncorrelated and therefore act as two different samples.

There are many variants of the percolation problem. For instance, one could also consider the case in which pores are all open and bonds or links are open or closed with a certain probability. While the first problem is denoted "site percolation", the second one receives the name of "bond percolation". One can be also interested in the dynamics of the percolation, i.e., how a percolation cluster is invaded starting from a given site or a given boundary. Other variants are possible.

Many problems may be reformulated as percolation ones. For instance, if one considers the problems of avoiding the spreading of fire in a forest, one may be concerned with the problem of the minimum fraction of trees to be cut to avoid the percolation of the fire. A similar problem is that of the spreading of a disease: What is the maximum fraction of unvaccinated individuals in a population, given its networks of contacts, that still avoid the percolation of a disease?

Many examples concern computer science and the Internet. Indeed, the birth of the Internet is just a percolation problem: a worldwide network arose from the sequential adding of links among nodes (computers) [15]. The resilience of the Internet to failure can be formulated as a percolation phase transition problem since its degree distribution implies the presence of heavily connected hubs [4].

Many routing protocols, which make possible dynamical communication among mobile devices [16], are based on the percolation concepts [17,18]. In addition, the determination of the critical coverage of an area with sensors is essentially a percolation problem [19], with applications to healthcare topics.

When considering the communication level on the Internet, more complex networks and processes arise. An example is the network given by the links among web pages, and the difficulty (and profitability) of exploring it [20].

Indeed, one of the main issue of the Internet of Things is the choice between hierarchic/centralized approaches and distributed/peer-to-peer ones [21]. From one side, the distributed approach is more scalable, but it requires special techniques for insuring the reachability of each node, the coverage of the whole domain, and the optimization of communications among different nodes. From the other side, a distributed approach implies the cooperation of the participating nodes. While this cooperation can be taken for granted when all sensors/devices belong to the same organization, it can break for heterogeneous devices or in the case of failure/hacking [22]. The dilemma between trusting and security and between cooperation and exploitation is at the basis of human behavior, and one can take inspiration from what we know about human heuristic, i.e., rules of thumb developed by humans to face similar problems [23–26] As a simple example, which is developed in the following, one can consider that the spreading of news or of rumors in a social network is quite similar to the spreading of a disease [27,28].

The percolation model can be extended in several ways. One possibility is the addition of nonlinear "reactions" on the nodes, as we discuss below. Another extension is that of considering only clusters whose sites are connected with more than $k$ links [29,30]. This "bootstrap" percolation [31] is strongly related to the robustness and resilience of a network structure.

The possibility of extending the basic model by introducing "nonlinear" processes on nodes are endless, and may be collectively indicated with the word "cellular automata" (see the proceedings of the *Cellular Automata* conferences for some interdisciplinary examples of this concept [32–40]).

Percolation is usually studied on static networks, however, if one thinks of the process of adding nodes to a growing cluster or of joining clusters together, one has the the view of a dynamic growing process. Growing networks exhibit properties quite different from those of static ones [2], and the relative theory and processes [3] can be applied to the Internet [4].

Percolation problems are linked to opinion or "spin" problems, such as the Ising or Potts models. In these models, the local stochastic dynamics induces an alignment (or counter alignment) of the state variables with those in the neighborhood of any given site. The main difference with the generic percolation problem is the presence in the latter of "absorbing states", i.e. a state of the whole network that cannot be abandoned, such as the global healthy state for epidemic spreading (in the absence of spontaneous appearance of a disease). Opinion models can therefore be considered as special cases of percolation ones.

The main ingredients of this generalized percolation problems are therefore the node dynamics and the connection network. We examine several examples in which the dynamics ranges from a simple "ferromagnetic" dynamics, i.e., a node tends to align to neighboring ones, to more complex dynamics, and the role of the topology in the global behavior.

There is also an interplay between epidemic and opinion dynamics: the actual spreading capacity of a disease among humans strongly depends on the perception of the risk of being infected and the consequent avoidance of dangerous contacts/practices and the visiting of crowded places. However, the effectiveness of this cognitive add-on crucially depends on the overlap between the contact network and the information one.

One of the interesting applications of these considerations to the computer world is in the field of automatic trading. We can abstract the problem thinking of nodes that process information and distribute if to the connected (neighboring) ones. A tainted or false opinion can spread or percolate

inside the network. However, since in general information is processed in a redundant way, it may happen that the tainted information spontaneously disappears (self-healing).

Nodes can check the reliability of the received information by contacting central repositories or conducting other actions, but this is a costly action in terms of time, which is the currency of automatic trading. It is therefore important to evaluate the risk of propagating an infection (considering the self-healing capacity of the network) vs. the cost of checking. This is very reminiscent of what happens in human networks, and similar strategies can be ported to the computer world. In the following, we examine the problem of quickly detecting the percolation cluster or the percolation threshold in various problems.

Finally, one may be interested in evaluating the spreading of some news, an error or a virus and the control of this spreading, possibly inside a limited region. One can consider two replicas of the same system, and slightly perturb one of them. The dynamics of the difference among them defines again a percolation problem, where the synchronized state is absorbing. Therefore, it is evident how synchronization and spreading characteristics are related, and also how the control of the spreading is connected to the problem of finding the fastest or most efficient way of synchronizing the two replicas, i.e., a targeted percolation.

In Section 2, we give a mathematical definition of the percolation problem and its connection with Monte Carlo implementation of equilibrium systems. In Section 3, we briefly discuss the role of absorbing states and non-equilibrium situations. In Section 4, we discuss the influence of network topology, in particular the "small world effect". Section 5 is dedicated to criticality and self-organized criticality, followed by risk perception in Section 6. Chaos, synchronization and control are discussed in Section 7 and the paper ends with some conclusions and comments.

## 2. Percolation and Related Models

Let us consider a network composed by $N$ nodes $i = 1, \ldots, N$. The links between nodes are defined by means of an adjacency matrix $a_{ij}$, $i, j = 1, \ldots, N$, and $a_{ij} = 1$ if there is a link from node $j$ to node $i$ and zero otherwise. The network needs not to be symmetric, and a node is generally connected to itself. One can also define multiple connection networks among the same nodes (multigraph) with different adjacency matrices $a_{ij}^{(1)}, a_{ij}^{(2)}, \ldots$.

We denote by $k_i = \sum_j a_{ij}$ the input connectivity of node $i$. The state of the node $i$ at time $t$ is denoted $s_i(t)$. It takes a value in a discrete set $s_i(t) \in \{0, 1, \ldots, q-1\}$, generally Boolean, $q = 2$. We also denote by

$$\mathcal{N}_i(\mathbf{s}(t)) = \left\{ s_j(t) \right\}_{a_{ij}=1} \tag{1}$$

the configuration at time $t$ of input nodes for node $i$, i.e., its *neighborhood*.

The state of the node $s_i(t)$ depends on the previous states of input nodes, and on some random number $r_i(t)$

$$s_i(t) = f\left(\mathcal{N}_i(t-1); r_i(t)\right). \tag{2}$$

For instance, the simple site percolation problem is defined by

$$s_i(t) = s_i(t-1) \vee [p < r_i] \left[ \sum_j a_{ij} s_j(t-1) > 0 \right], \tag{3}$$

where $\vee$ is the logical OR; $[\cdot]$ is the truth function that takes value 1 if $\cdot$ is true and zero otherwise; $s_i = 0$ denotes a healthy/dry node; and $s_i = 1$ denotes an infected/wet one. The logic of the previous equation is that an infected node stays infected, while a healthy one can become infected with probability $p$ if at least one of the connected nodes is infected. Notice that the random numbers do not change in time, thus, for a given value of $p$ and a given starting configuration (say one infected node), the clusters of infected nodes grows for some time and then freezes.

In the following, we neglect to indicate the time/space dependence of the random numbers $r_i(t)$, and the fact that the function $f$ of Equation (2) may depend on them.

Actually, the standard percolation problems [10] do not include the "time" dimension, which is instead included in the "directed" formulation [41]. As noted, it is however possible to unify the two definitions: if the random numbers are renewed at each time step, it is a directed percolation problem, while, in the contrary case, it is a static percolation one.

Let us denote by $s(t) = \{s_i(t)\}$ the state of the whole configuration at time $t$, which evolves following a stochastic or deterministic rule. In the case of a stochastic evolution, one needs a certain amount of random numbers $r(t) = \{r_i(t)\}$, uniformly distributed in the interval $[0, 1)$. These numbers can also not depend on $t$, as in the case of standard percolation. They can be seen as a property of the space-time lattice, i.e., a "quenched" random field.

In general, the time evolution of $s$ depends on a certain number of *control* parameters that we denote with $p$, as in Equation (3). These control parameters define the local transition probabilities $\tau(s_i'|\mathcal{N}_i(s); p)$, where $\mathcal{N}_i$ is the state of the neighborhood of site $i$, Equation (1), and $s_i'$ denotes the new state of that site. Clearly, $\sum_{s'} \tau(s_i'|\mathcal{N}_i(s); p) = 1$.

We restrict our exposition to the Boolean case $s_i \in \{0, 1\}$. In this case, the evolution of the system is given by the parallel application of a local rule

$$s_i(t+1) = [r_i < \tau(1|\mathcal{N}_i(s(t)); p)].$$

Given the random field $r_i(t)$, the dynamics is deterministic, but the final state depends on the field itself. If all quantities $\tau$ take the value of either zero or one, the evolution of the system is deterministic and does not depend on the field $r_i(t)$.

We introduce also a set of observables $A(s)$, for instance the "density" or "magnetization" $m(s) = (1/N) \sum_i s_i$. The time average of an observable is

$$\overline{A} = \frac{1}{T} \sum_{t=t_0}^{t_0+T} A(s(t)),$$

where $t_0$ denotes the transient interval. In principle, this quantity depends on the initial configuration and on the random field. In practice, in many cases, after a long enough transient $t_0$ (which depends on the system size $N$) and for a long enough average time $T$, the average values does not depend on $t_0$, $T$ or the field. In such cases, the system is said to be "self-averaging". This is not the case, for instance, for spin-glasses, introduced below.

Averaging over the "disorder" $r(t)$, one can define a probability distribution $P(s, t)$, whose time evolution is

$$P(s, t+1) = \sum_{s'} M(s|s')P(s, t), \tag{4}$$

where $M$ defines a Markov chain, and is given by

$$M(s'|s) = \prod_i \tau(s_i'|\mathcal{N}_i(s); p). \tag{5}$$

Notice that $P(s, t)$ depends in principle on the initial distribution $P(s, 0)$, which can be a delta function centered on some configuration $s_0$.

The ensemble average value of an observable can be computed over the probability distribution

$$\langle A \rangle = \sum_s A(s)P(s).$$

For self-averaging systems, $\overline{A}$ coincides with $\langle A \rangle$.

If the asymptotic probability distribution for any given initial state coincide, the system is said to be ergodic. In a phase transition, one has the breaking of ergodicity, so that the probability distribution depends on the initial state of the system.

For the standard percolation and infection problems, one assumes that the spreading process occurs independently on each link and/or on each node. We can generalize the problem by formulating it as a cellular automaton model [42,43], in which the function $f$ of Equation (2) can be arbitrary. In this way, we can "include" into the same formulation many other interesting problems.

One of the first generalizations is that of "diluting" the application of any rule, i.e., replacing Equation (2) with

$$s_i(t) = [q < r]f\left(\mathcal{N}_i(t-1)\right) + (1 - [q < r])s_i(t-1).$$

or, equivalently

$$\tau(s|\mathcal{N}) = \begin{cases} f\left(\mathcal{N}_i(t-1)\right) & \text{with probability } q \\ s & \text{otherwise} \end{cases}$$

The previous equation implies that, on average, only a fraction $q$ of nodes are updated, while the others keep their values. In the limit $q \to 0$, one or none of nodes are updated, and therefore one gets time-continuous dynamics.

Another extension is that of getting the transition probabilities from the Monte Carlo realization of an equilibrium model, i.e., from an Hamiltonian (energy) term. Let us illustrate this concept using the famous Ising model [44,45]. In this model, each node is associated to a spin $\sigma_i = -1, 1$ and the energy of a configuration $\sigma$ is given by

$$\mathcal{H}(\sigma) = -\sum_i J\sigma_i h_i,$$

where $J$ denotes the coupling and $h_i$ is the local magnetic field, due to neighbors

$$h_i = \sum_{\sigma_j \in \mathcal{N}_i(\sigma)} \sigma_j = \sum_j a_{ij}\sigma_j. \tag{6}$$

The maximum entropy condition gives the equilibrium (asymptotic) distribution

$$P(\sigma) = \frac{1}{\mathcal{Z}} \exp\left(-\frac{\mathcal{H}(\sigma)}{T}\right),$$

where $T$ is the temperature in energy units and $\mathcal{Z}$ is the normalization constant (partition function). The actual control parameter is the ratio $J/T$.

The idea is that, for strong coupling $J$ or low temperature $T$, the probability is peaked over configurations with many aligned spins (if $J > 0$, counter-aligned of $J < 0$) and average magnetization $|m| > 0$, while on the contrary the distribution is almost flat and $m \simeq 0$. The low-temperature phase corresponds to a percolating phase, where there is at least one big cluster spanning the entire system: compare Figure 1 with Figure 2. In any case, the correlation function

$$g(r) = \langle \sigma_i \sigma_{i+r} \rangle - \langle \sigma_i \rangle^2$$

is an exponentially decreasing function of $r$,

$$g(r) \propto \exp\left(-\frac{r}{\xi}\right),$$

where $\xi$ is the correlation length.

Near the critical temperature, for $T \simeq T_c$, there are clusters of all sizes, so the average magnetization is at the boundary of the zero-phase and the correlation length diverges, while the correlation function takes a power-law shape (see Figure 2).

Actually, numerical computations are not based on on computing the probability distribution over all possible configurations, rather in generating a Markovian path through configurations via the Monte Carlo technique, such that on average the trajectory spends time visiting a configuration proportional to $P(\sigma)$. There are many Monte Carlo recipes, mostly based on the detailed balance principle

$$\frac{M(s|s')}{M(s'|s)} = \exp\left(\frac{\mathcal{H}(s') - \mathcal{H}(s)}{T}\right),$$

and they can be interpreted as a diluted CA rule. For instance, the heat-bath recipe gives the following transition probabilities of getting spin $\sigma_i$ at time $t + 1$,

$$\tau(\sigma|\mathcal{N}_i) = \frac{1}{2}\left(1 + \sigma_i \tanh\left(\frac{Jh_i}{T}\right)\right),$$

where $h$ is given by Equation (6).

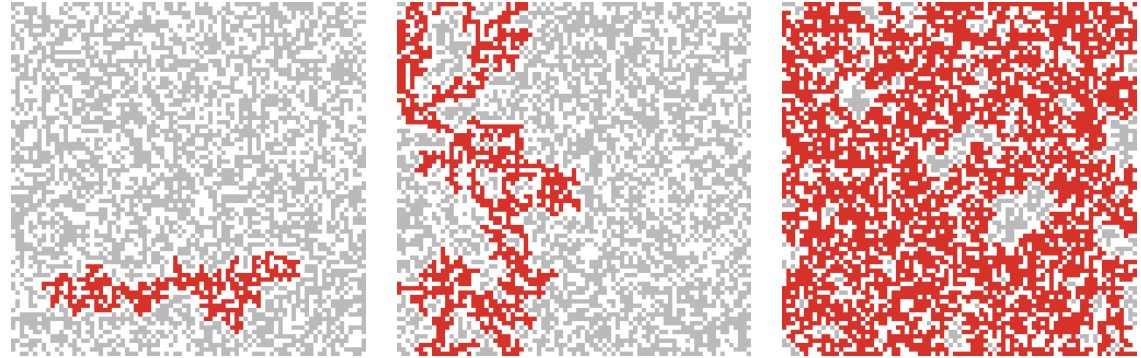

**Figure 1.** Typical patterns of the percolation model, where the largest component is marked in red: (**Left**) $p < p_c$; (**Middle**) $p = p_c$, notice the appearance of a percolating cluster; and (**Right**) $p > p_c$.

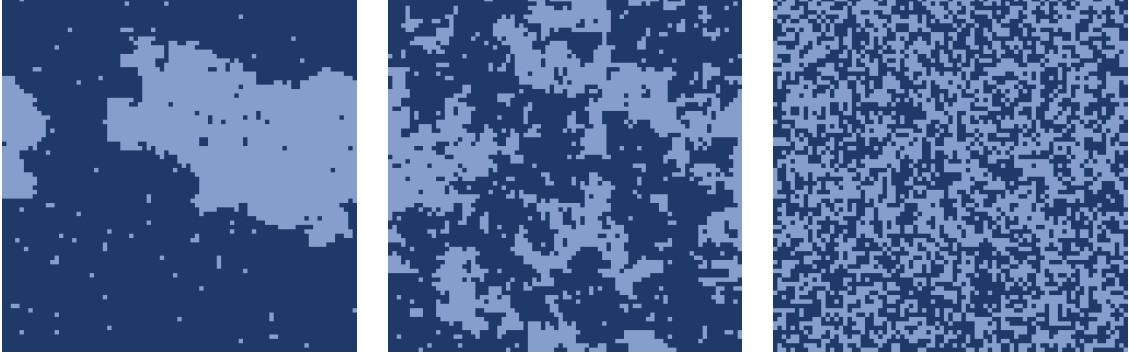

**Figure 2.** Typical patterns of the Ising model: (**Left**) $T < T_c$; (**Middle**) $T = T_c$, notice the appearance of a percolating cluster; and (**Right**) $T > T_c$.

In Monte Carlo computations, it is possible (using an appropriate choice of the sites to be updated and of the flipping probability [46]) to update more than one spin at the time, but this technique is generally implemented in the "diluted" sense of just one update per time step. However, one can also consider the fully parallel case, which may be better suited for extending the magnetic analogy to the dynamics of opinions [47].

In this view, a spin "up", $\sigma_i = 1$, is considered equivalent to a person holding a given opinion, while the opposite spin "down", $\sigma_i = -1$, corresponds to the opposite opinion [48]. The magnetized

phase corresponds to the emergence of well-defined majority, and the model may be extended studying the influence of stubborn people (leaders), as in the Social Impact Theory [49], different dynamics of opinion evolution [50], etc.

In most models, the influence of neighbors act independently of each other, essentially obeying a superposition principle. However, social experiments [51,52] show that the presence of a strong majority can affect the social orientation more that the separate cumulative effects, i.e., impose a *social norm*. In the language of magnetic systems, this effect can be translated into a "plaquette" term, i.e., a term that depends on the presence of a certain number of aligned spins in the neighborhood. It can be easily verified that this influence can be translated in terms involving the products of more than two spins or, equivalently, in powers of the local magnetic field [53]. Thus, one can consider for instance Hamiltonians of the form

$$\mathcal{H}(\sigma) = -\sum_i \sigma_i \left( Jh_i + Zh_i^2 + Wh_i^3 \ldots \right).$$

The effect of plaquette terms becomes important in conjunction with the change in topology, as illustrated in Section 4. In pure percolation models, similar terms constitutes the bootstrap [31] or *k*-core [29,30] percolation models. In some sense, these models are analogous to the Game of Life [54,55], without the possibility of giving birth to new cells.

Another important generalization is that of having more than two symbols/opinions. This generalization is called Potts model [56], which coincides with the Ising one for just two symbols. Its Hamiltonian is simply defined as

$$\mathcal{H}(s) = \sum_{ij} Ja_{ij}[s_i = s_j],$$

where the $[\cdot]$ function acts as a Kronecker delta, and $s_i \in \{0, 1, \ldots, q-1\}$. This model allows better defining the clusters, i.e., groups of connected sites having the same value. In the Ising model, it may happen just for coincidence that two connected neighbors take the same value. In this model, a "real" cluster is a subgroup of the set of connected, same-value sites, and can be identified by cutting, inside such a group, links with a probability $\exp(-J/T)$. In this way, even below the critical temperature $T_c$ where there is a giant component of equal-value spins, clusters result to be finite, and indeed their distribution diverges only at $T_c$. This definition of a cluster is the basis for fast parallel simulations near the critical point [46,57].

In the limit of large number of states, the clusters of same-value state in the Potts model actually correspond to real clusters. Indeed, the Potts model can be used as a data clustering algorithm [58]. It is interesting that the transition becomes sharper and sharper, as the number *m* of states increases, and for $q > 4$ it becomes discontinuous (first order), as happens for the *k*-core percolation problems.

It can be proved that the limit $q \to 1$ of the standard Potts model is equivalent to bond percolation [59] and, by adding plaquette terms, to the site percolation problem [60].

The phase transition can be described as a bifurcation, exploiting the mean-field approximation. Disregarding the correlations among neighboring spins, one can assume that the local magnetic field *h* coincides with the average magnetization *m* times the input connectivity *k* (assumed constant), obtaining an evolution equation for *m*

$$m(t+1) = \tanh\left(\frac{Jkm}{T}\right). \tag{7}$$

By varying *T* or *J*, one can have a bifurcation from one to three fixed points, i.e., from $m(\infty) = 0$ to $|m(\infty)| > 0$, as shown in Figure 3.

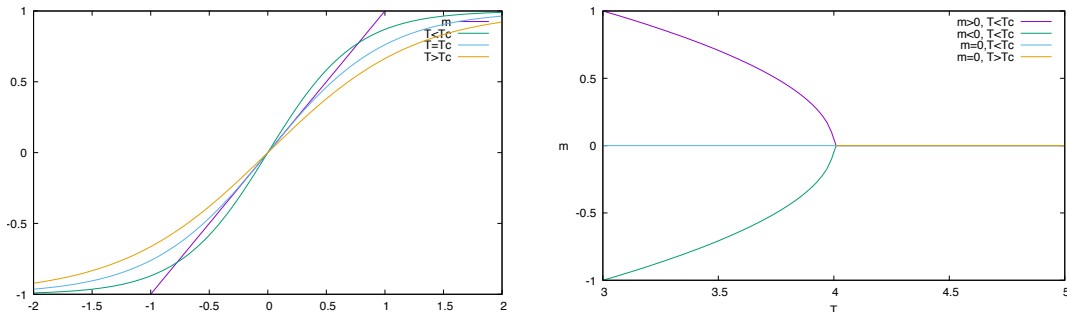

**Figure 3.** The bifurcation scenario of Equation (7) by varying $T$: (**Left**) the asymptotic values of $m(\infty)$ are given by the intersection of $f = m$ with $f = \tanh(Jkm/T)$ in the left plot; and (**Right**) location of $m(\infty)$ as a function of $T$.

Alternatively, one can assume that the energy is given by $U = -NkJm^2/2$. The probability of finding a spin up (1), given a magnetization $m$, is $P(1) = (1+m)/2$, and $P(-1) = (1-m)/2$. The entropy is thus given by

$$S = -N\left[\left(\frac{1+m}{2}\right)\ln\left(\frac{1+m}{2}\right) + \left(\frac{1-m}{2}\right)\ln\left(\frac{1-m}{2}\right)\right],$$

and the free energy is $F = U - TS$. The magnetization is given by the minimum of $F$ with respect to $m$, obtaining

$$Jkm = \frac{T}{2}\ln\left(\frac{1+m}{1-m}\right),$$

which gives the asymptotic value of Equation (7). By varying $T$, one gets the view of the bifurcation of Figure 3 as a variation of the minima of $F$, as shown in Figure 4.

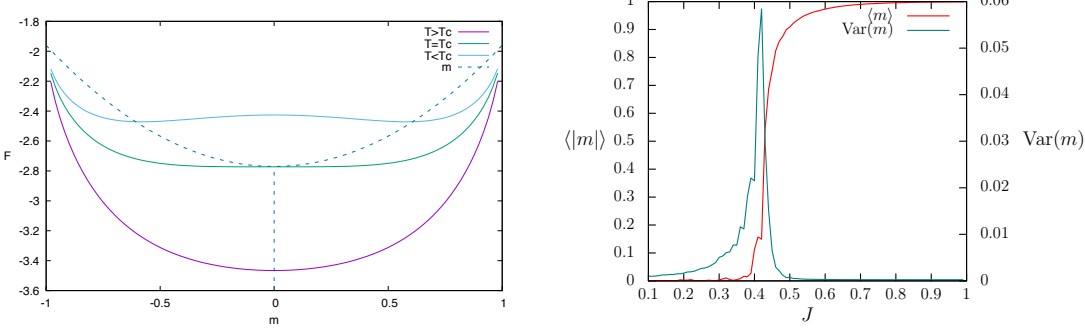

**Figure 4.** (**Left**) Plot of the free energy $F$ as function of the magnetization $m$ and for various temperatures $T$. The location of the minima, $m(\infty)$ is marked by the dashed line. (**Right**) Numerical simulations for $N = 20 \times 20$ of the Ising model, by varying $J$ and $T = 1$, average over 4000 time step, after a transient of $4 \times 10^4$ steps.

More details about phase transition in percolation, Ising and cellular automata models can be found in [61].

Finally, a whole new phenomenology arises in the case of disordered links (for instance, $a_{ij} = \pm 1$), models that go under the name of "spin glasses" [62–66]. They are strongly related to attractor neural networks [67]. For high temperatures, "standard" disordered configurations are present. However, there are now many low-energy states, separated by high energy barriers. They originate by frustrations: spins are unable to find a configuration that satisfies all links. This means that the systems takes a long time exploring the "energy valley", jumping over energy barriers, and in practice they never reach equilibrium.

The same definition of an order parameter is problematic, since one cannot simply "count" the number of up or down spins. The idea for identifying an more ordered phase is that of using a *replica* $\eta$ of the system $\sigma$ (with the same random coupling but starting from a different configuration) and computing the average overlap $q$ or distance $1 - q$ between them

$$q = \frac{1}{2}\left(1 + \frac{1}{N}\sum_i \sigma_i \eta_i\right) = \frac{1}{N}\sum_i [\sigma_i = \eta_i].$$

Both overlaps are different from zero in the low-temperature phase (with continuous phase transitions). The overlap concept is also used when speaking of control and risk perception.

## 3. Absorbing States and Non-Equilibrium Processes

In a true percolation problem (for instance, in epidemic spreading), once the wetting (or the infection) is stopped, it can never restart again. Let us assume that $s_i = 0$ denotes the dry/healthy state and $s_i = 1$ the wet/infected one. Denoting by $s = 0$ (1) the state in which all sites have value 0 (1), we have that the state $s = 0$ cannot be abandoned, i.e., $M(0|0) = 1$ and $M(s|0) = 0$, for $s \neq 0$. This behavior cannot be obtained by a finite Hamiltonian, Equation (5), so we have to abandon the equilibrium concept.

The simplest model is probably the Domany–Kinzel (DK) one [42]. In this model, the state of a site depends symmetrically on that of two neighbors in the previous time step; without loss of generality, one can say that

$$s_i(t+1) = f(s_i(t), s_{I+1}(t); r_i(t)) = [r_i(t) < \tau(1|s_i + s_i + 1)].$$

For the DK model, one has $\tau(1|0) = 0$, $\tau(1|1) = p$ and $\tau(1|2) = q$, where $p$ and $q$ are control parameters. One can recover the 1+1 direct site percolation model for $p = q$ and the bond direct percolation model for $q = p(2 - p)$ [43]. For $p = 1, q = 1/2$ one has the parallel Ising model at zero temperature.

The order parameter $c$ is the average asymptotic number of "wet" sites

$$c = \frac{1}{N}\sum_i s_i.$$

The phase diagram of the DK model is reported in Figure 5. The simplest mean-field approximation consists in disregarding all correlations, thus the probability of observing a wet/infected site is just $c$ and its evolution equation reads

$$c' = 2pc + (q - 2p)c^2,$$

from which one can get the two asymptotic values of the density $c_1^* = 0$ (absorbing state) and $c_2^* = 2p/(2p - q)$ (active state). The stability switch among the two values occurs for $p = 1/2$, quite different from what happens for a real lattice (Figure 5). The general scenario is however well described by the mean-field approximation, with a continuous transition between the two values that becomes more abrupt with increasing $q$. Indeed, for $q = 1$, a compact patch of zeros or ones always stays compact, and therefore the dynamics is dominated by an annihilating random walk of the boundaries (two boundaries annihilate when the patch between them disappears). Due to the symmetry, this transition occurs strictly for $p = 1/2$ (also for lattices) and it is discontinuous (in the asymptotic limit either the lattice is formed by zeros or by ones). By keeping in consideration larger correlations, one can get more accurate approximations, as shown in Figure 5 (left). Another interesting transitions occurs on the line $q = 0$, since in this case the evolution can be considered as the dilution of

a deterministic CA rule, denoted rule 90 in Wolfram notation [68]. This rule states that the future value of the state of a site is given by the *exclusive or* ($\oplus$) of the state of the two neighbors

$$s_i(t) = s_{i-1}(t-1) \oplus s_{i+1}(t-1).$$

Since this rule only depends on the neighbors, the system actually decouples (for even $N$) in two non-interacting sublattices, as shown in Figure 6. Near the transition, one of the two sublattices will enter the absorbing state before the other, and in any case the probability that both have a surviving cluster in the same location is negligible, so that the dilution is simply given by a reduced probability of getting a one. Typical patterns of the surviving cluster near the transition are reported in Figure 7; one can see that, by increasing $q$, the compactness of clusters also increases.

Notice that, while the density value $c_1^* = 0$ corresponds to a unique state $s_i = 0$, the other value $c_2^*$ arises as an observable computed over a wide distribution of states. The absorbing state can be seen as a configuration at a negative infinite energy, thus it cannot be left whatever the temperature. Since there is always a probability of jumping from any (finite) configuration into the absorbing state, the phase separation occurs only in the limit $N \to \infty$, which should be taken before the time limit $t \to \infty$.

The implementation of the DK model can be done using one or more random number per sites [69]. Assuming just a random number, it can be expressed as

$$s_i(t) = [r_i(t) < p](s_{i-1}(t) \oplus s_{i+1}(t)) + [r_i(t) < q](s_{i-1}(t)s_{i+1}(t)).$$

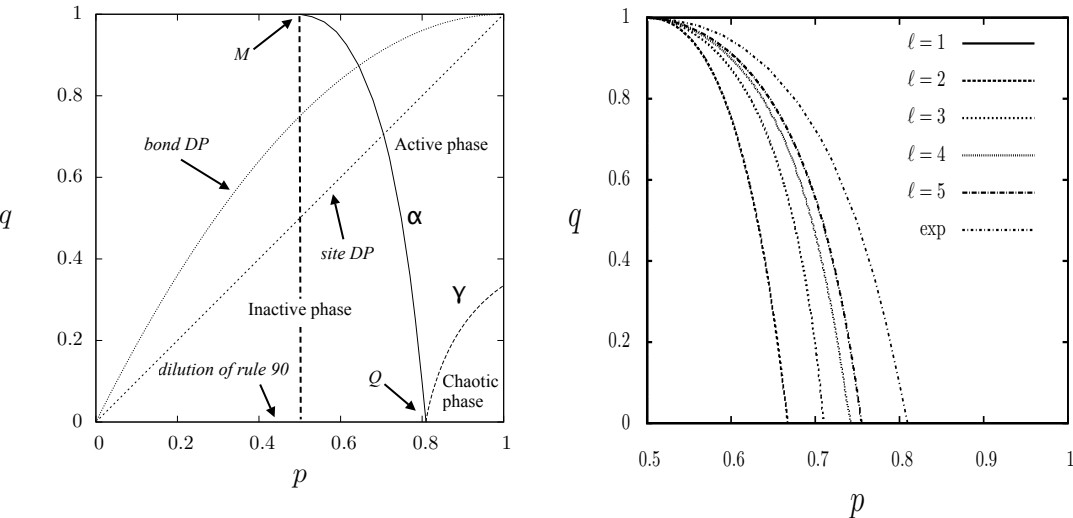

**Figure 5.** (**Left**) The phase diagram of the Domany–Kinkel model, where $\alpha$ marks the density transition and $\gamma$ the damage transition. The dashed line marks the transition line for the simplest mean-field approximation. (**Right**) Several improved mean-field approximations. Figure from [61].

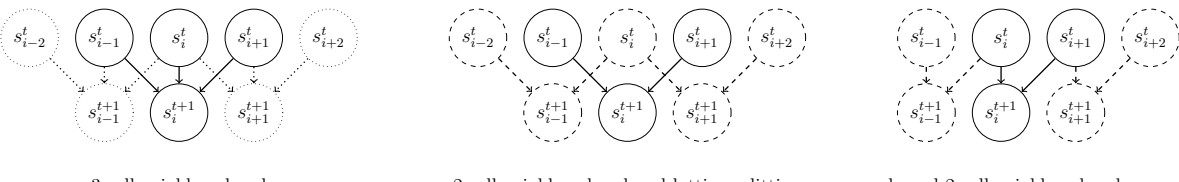

**Figure 6.** Decoupling of the lattice and skewed neighborhood.

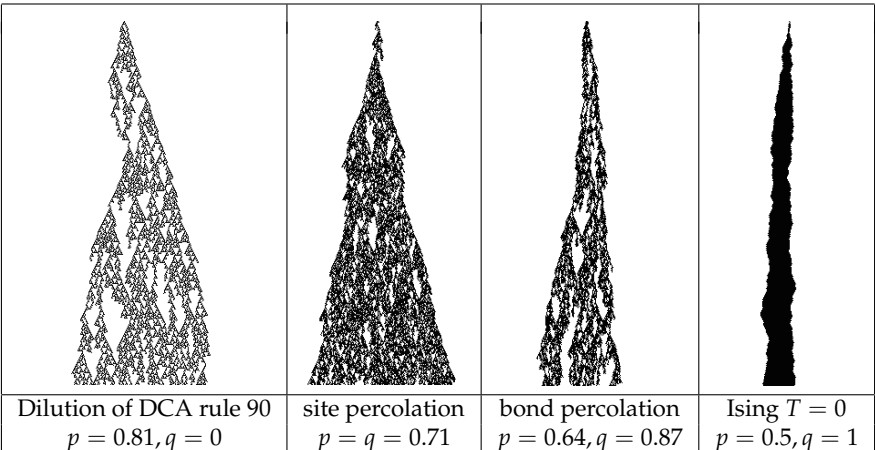

| Dilution of DCA rule 90 | site percolation | bond percolation | Ising $T = 0$ |
|---|---|---|---|
| $p = 0.81, q = 0$ | $p = q = 0.71$ | $p = 0.64, q = 0.87$ | $p = 0.5, q = 1$ |

**Figure 7.** Typical patterns of the DK model. Space runs horizontally and time vertically, from top to bottom. Figures from [61].

One can think of extracting the set of all $\{r_i(t)\}$ once for the whole simulation, as if they were a quenched random field (for instance, differences in treating data in a processor network). An interesting question is whether the final distribution depends on the initial one, or just on the choice of the quenched field. In other words, given the set of $\{r_i(t)\}$, the automata becomes deterministic, and one may wonder whether a small difference in the initial configuration would spread (percolate) in the network. This question is analogous to the problem of diffusion of errors in distributed processing of information. Numerical simulation and analytical approaches [70] show that this "chaotic" phase lies within the active one (see Figure 5, right). The rest of the active phase can be denoted with the term "disordered", since it is

In opinion models, it is more common to consider the symmetric role of the two sides. Let us generalize the DK model, assuming that both the homogeneous zero and one states are absorbing, the so-called Bagnoli-Boccara-Rechtman (BBR) model [71], that is a three-input cellular automata. It is a totalistic automaton, meaning the transition probability depends on the sum $S$ of the states in the neighborhood, with $0 \leq S \leq 3$. The BBR transition probabilities $\tau(1|S)$ are $\tau(1|0) = 0$, $\tau(1|1) = p_1$, $\tau(1|2) = p_2$, $\tau(1|3) = 1$

As can be seen in Figure 8 (right), we have here, for high-$p_1$ and low-$p_2$ values, two DP transitions reminiscent of the DK model. These two lines meet at about $p_1 = p_4 = 0.5$ ($p_1 = 1 - p_2 = 1/3$ in the mean-field approximation). In this point the universality class changes to that of parity conservation [72]. In the low-$p_1$, high-$p_2$ part of the diagram, we have a first-order transition: the two absorbing states are stable (as predicted by the mean-field analysis) and we can investigate the nature of an hysteresis cycle. To do that, we have to remove the absorbing characteristic of the states 0 and 1. We do this by imposing that $w = \tau(1|0)$ is small but different from zero, so that in principle the system does not more exhibit a true phase transition. Indeed, that states with high or low values of the density are now metastable, so we have to tune the simulation time with the value of $w$. This tuning is however not critical: for a large range of values of simulations times, we obtain an hysteresis diagram similar to that of the inset of Figure 8 (right).

In the BBR diagram, we also have, beyond the absorbing phases, an active phase where the asymptotic state is given by a superposition of many configurations, with a broad probability distribution. As shown in Figure 8 (left), the active zone can be further divided into a disordered one, where an initial damage is always re-adsorbed and the final state only depends on the random field $r_i(t)$, and a chaotic one, near the corner $p_1 = 1$, $p_2 = 0$, similarly to what happens in the DK model.

One can think of the BBR model as an opinion model in the presence of social norms: if the local majority is larger than a given threshold, such as in the Asch experiment [51,52], the alignment with the majority is complete. To explore the consequences of this assumption, let us increase the size of the neighborhood [73,74]. Let us denote by $k$ the number of connected neighbors, $k = \sum_j a_{ij}$, and denote

by $S_i$ the sum of the value of neighbors (the equivalent of the local field), $S_i = \sum_j a_{ij}s_j$. We generalize the BBR model on the symmetry line $p - 2 = 1 - p_1$ by establishing a threshold $Q$ such that

$$\tau(1|S) = \begin{cases} 0 & \text{if } S \leq Q \\ 1 & \text{if } S \geq k - Q \\ \frac{1}{1+\exp(-J(2S-k))} & \text{otherwise} \end{cases} \qquad (8)$$

For $J < 0$, the system shows frustration, since a spins tends to anti-align with the local majority if this is marginal, but adheres to it when it is strong enough. We have therefore a sort of "reasonable contrarians". This frustrated behavior causes the appearance, in the chaotic phase, of cluster of aligned nodes, the "triangles" in Figure 9. For comparison, the absorbing phase boundary for small $p_1$ corresponds to positive values of $J$, the point $p_1 = 0.5$ to $J = 0$ and the chaotic zone to large negative values of $J$.

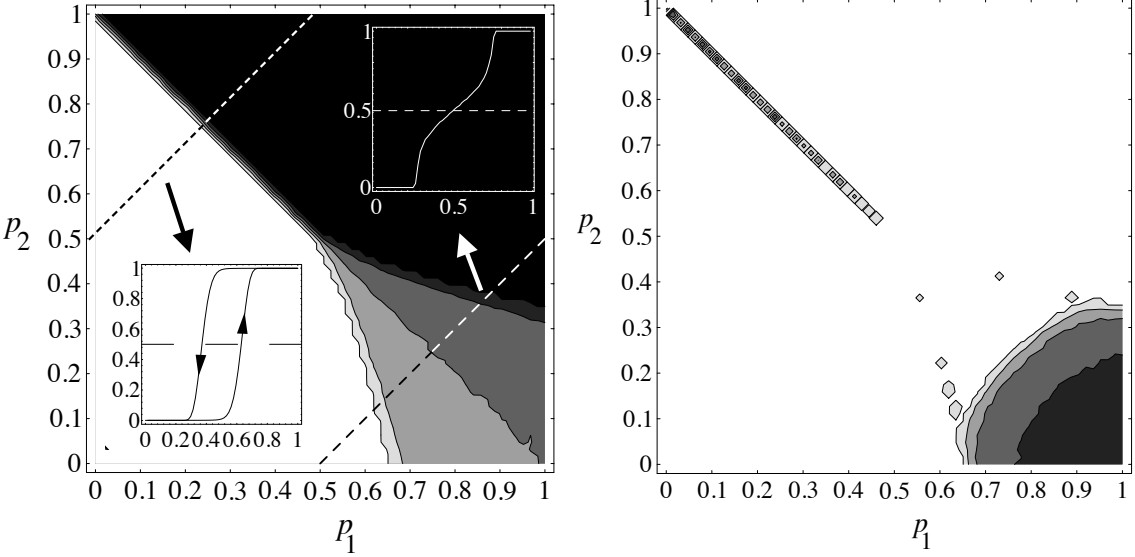

**Figure 8.** Phase transition diagrams of the BBR model (color code: white = 0, black = 1): (**Left**) Numerical phase diagram of the density, in the inset the variation of the density when cutting the phase diagram; the hysteresis inset at bottom right is obtained by setting $w = 10^{-4}$, $T = 500$. (**Right**) Chaotic phase. The sparse dots mark the boundary separating the quiescent phases from the disordered one. Numerical simulations with $N = T = 10^4$. Figures from [71].

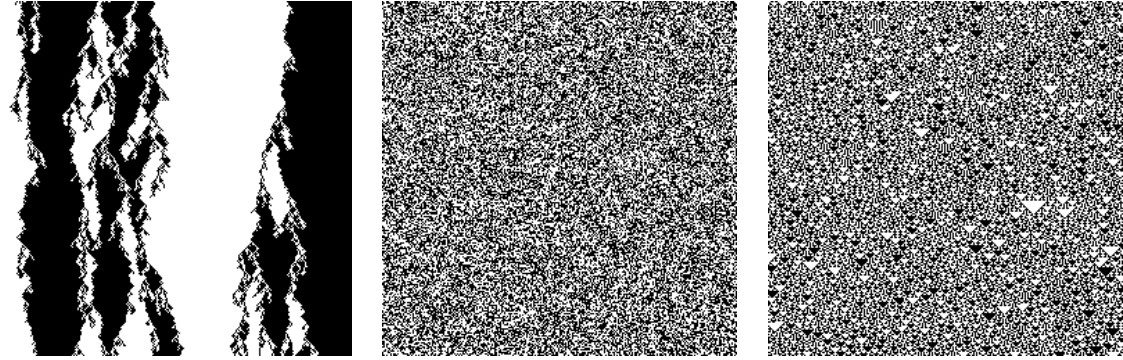

**Figure 9.** Typical space time patterns of the BBR model for $p_2 = 1 - p_1$) with $N = T = 256$. Time increases downward: (**Left**) competition between quiescent phases $p_1 = 0.1$; (**Middle**) disordered phase $p_1 = 0.5$; and (**Right**) chaotic phase $p_1 = 0.6$. Figures from [73].

By increasing $k$, one can see larger patches in the chaotic phase, and the appearance of a new phase at the boundary between the disordered and the chaotic one, where the patterns become insensitive not only to the initial conditions, but also to the variation of parameters. The reason for this behavior is that, if the density is near 0.5, $S \simeq k/2$ and therefore the probability of Equation (8) becomes $1/2$, independently of $J$, and this locks the evolution [73].

## 4. Network Influence and Small World Transitions

Up to now, we have treated either regular lattices or the mean-field approach, which implies the absence of correlations and therefore is equivalent to reshuffling the sites in a lattice at each time step. Actually, for quenched disordered connections, the prediction of the mean-field approximation are also generally quite good. It is possible to investigate what happens when we partially disorder an otherwise regular lattice. An interesting procedure was proposed by Watts and Strogatz [75], and consists in replacing a fraction $p$ of neighboring connections with random links (see Figure 10).

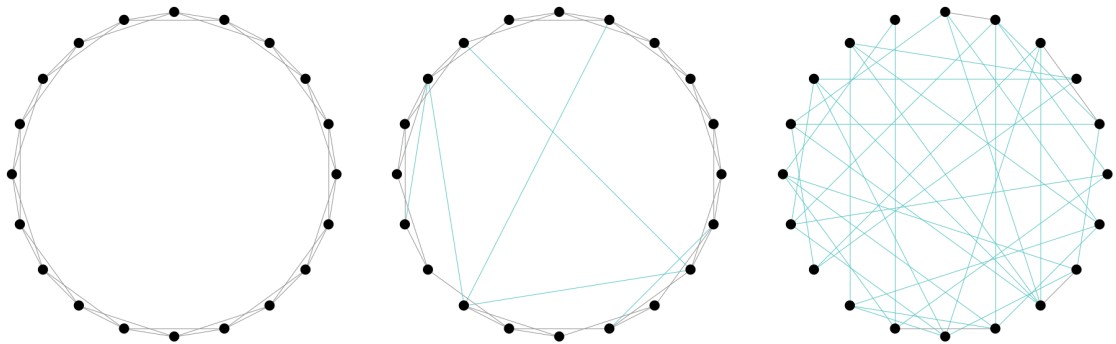

**Figure 10.** Rewiring in the Watts–Strogatz model [75]. (**Left**) $p = 0$; (**Middle**) $p = 0.2$; and (**Right**) $p = 0.9$.

In general, what happens with this procedure is that global observables like the density of "wet" sites assume the mean-field values also for moderate values of $p$, while local observables such as the probability of having small-length loops are still similar to the value they take in the regular lattice. For this, it is called the "small world" effect" if we consider diffusion of people or percolation of a disease among humans, both are quite slow processes in a regular lattice (or in a disordered lattice with local connections), but, as soon as long-range travel become possible (by boat or by airplane), the diffusion properties change, even if at the local level almost nothing has changed.

An interesting effect arises when the mean-field approximation gives chaotic oscillations, as happens in the long-range opinion model of the previous Section. The plot of the mean-field approximation of the rule of Equation (8) is reported in Figure 11 [74]. For sufficient large negative values of $J$ and large connectivity $k$, the map becomes chaotic. This chaotic character does not depend on the presence of absorbing states, and indeed they have been removed by inserting a small probability of opposing to the strong majority. Notice altogether that this chaos has nothing to do with the "microscopic" chaos already seen: the mean-field approximation refers to average quantities, so it implies a coherent oscillation of the whole, synchronized, network.

The simplest mean-field description of the model described in Equation (8) is given by

$$c' = \sum_{S=0}^{k} \binom{k}{S} c^w (1-c)^{k-S} \tau\left(1|S\right), \tag{9}$$

with $c' = c(t+1)$ and $c = c(t)$. In Figure 11, we show some graphs of this map by varying $J$ and $k$. The bifurcation digram of this map after varying $J$ is shown in Figure 12 (left).

The doubling bifurcation route to chaos ends at $J = J_c$. For $0 > J \geq J_2$ and $J_3 > J \geq 6$ there is only one attractor (blue, darker dots). For $J_2 > J \geq J_c$ there are two, one corresponding to the lower

branches that bifurcate up to $J_c$ (red, lighter dots), and the other one to the upper branches (blue, darker dots). For $J_c > J \geq J_3$, there are two chaotic attractors, one corresponding to the lower branches (blue, darker dots), and the other to the top branches (red, lighter dots). For every value of $J$, the dots are 64 iterates of the map after a transient of $10^3$ time steps. For values of $J$ with only one basin of attraction, the orbits do not depend on the initial average opinion $c(t = 0)$. For values of $J$ that correspond to two attractors, one of them is found with $c(0) = 0.1$, the other one with $c(0) = 0.9$.

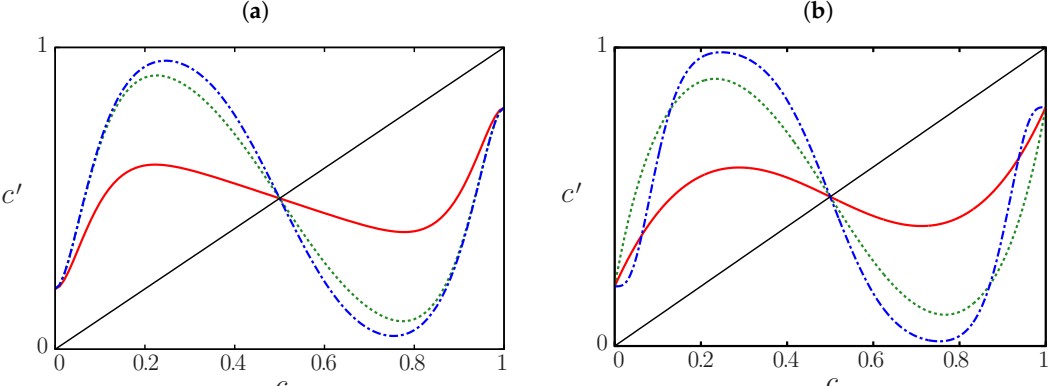

**Figure 11.** (**a**) Graphs of the mean field map corresponding to Equation (8) for different values of $J$ and $k = 20$. From bottom to top for $c < 1/2$, $J = -0.5$ (red, lower line), $J = -3.0$ (green, middle line), and $J = -6.0$ (blue, upper line). (**b**) Graphs of the mean field approximation for Equation (8) for different values of $k$ and $J = -6$. From bottom to top, for $c \sim 0.2$, $k = 4$ (red, lower line), $k = 10$ (green, middle line), and $k = 38$ (blue, upper line). Figures from [74].

By varying the long-range probability $p$, we observe the transition towards the mean-field behavior, as reported in Figure 13. This induces a stochastic bifurcation diagram by varying $p$ (Figure 12, right) that is quite similar to that obtained in the mean-field approximation by varying $J$ (Figure 12, left). For $p \lesssim p_0$, there are almost periodic orbits of period one and for $p_0 \lesssim p \lesssim p_1$ of period two. For $p_1 \lesssim p \lesssim p_2$, we find two attractors, one (in red, lighter) in the lower branches, and the other one (in blue, darker) in the top ones.

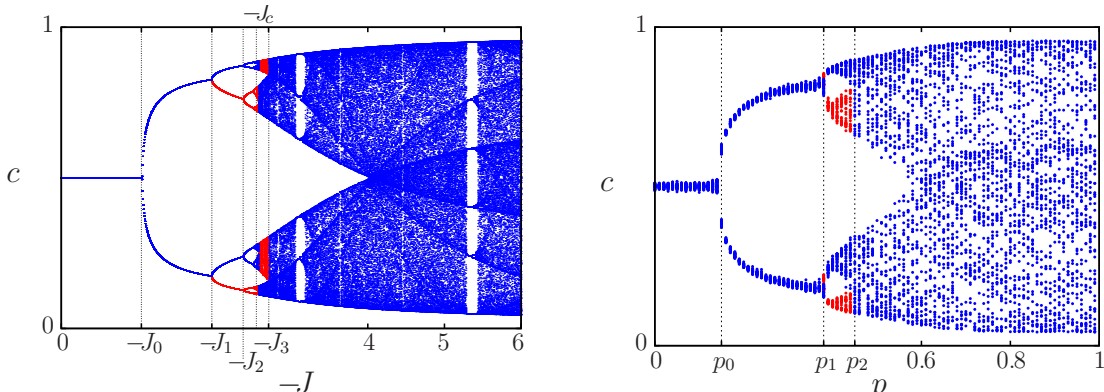

**Figure 12.** (**Left**) Bifurcation diagram of the mean-field map, Equation (9), by varying $J$; and (**Right**) small-world probabilistic bifurcation diagrams as functions of the long range probability $p$. The colors mark different attractors. Figures from [74].

A similar bifurcation diagram can be seen also in the parallel Ising model with plaquette terms (see [53]).

Finally, networks are much more than just regular lattices and disordered graphs. Many dynamics network show quite a different structure [3], without a well-defined scale, and in many cases there are many layers with different structures that interacts (multiplex networks) [76].

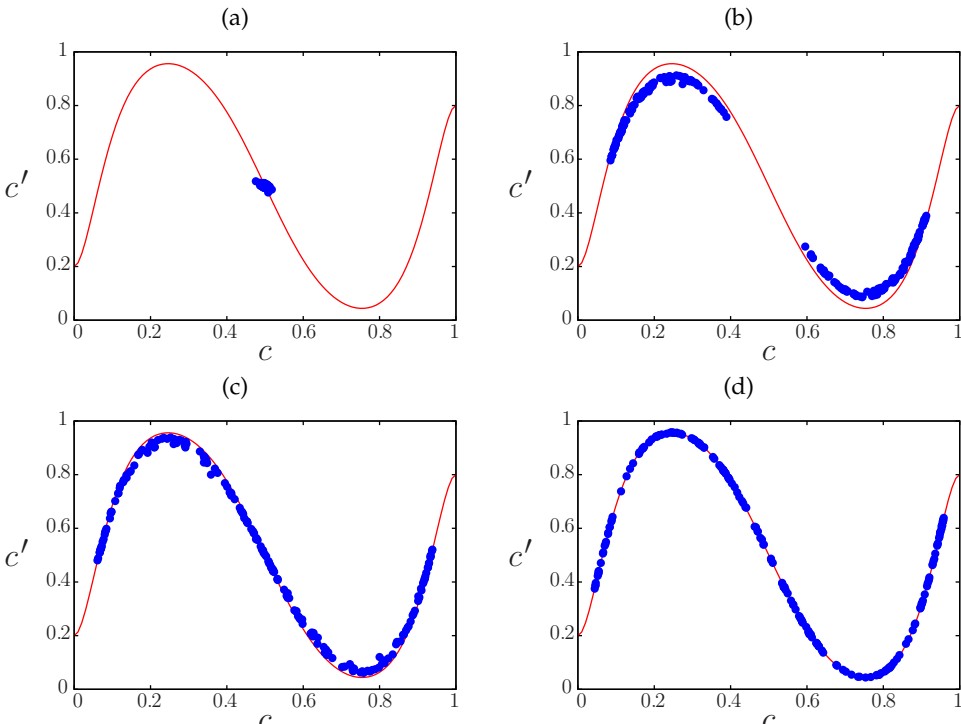

**Figure 13.** Return map of the average opinion $c$ on small-world networks for several values of the long-range connection probability $p$ with $J = -6$, $k = 20$, $N = 10^3$, and a transient of $10^3$ time steps. The following 200 iterations are shown as (blue, darker) dots. The (red, lighter) continuous curve is Equation (9). (**a**) $p = 0.0$, (**b**) $p = 0.5$, (**c**) $p = 0.6$, and (**d**) $p = 1.0$. Figures from [74].

## 5. Criticality and Self-Organized Criticality

The topic of phase transitions has fascinated physicists for many years, originally for computations related to high-energy physics and then to continuous phase transitions in statistical mechanics (Ising) models [77]. The idea is that, near the phase transition, the correlation length diverges, and clusters of all sizes are present, thus one can perform a coarse-graining (renormalisation) of the lattice, and exactly at the boundary of the phase transition this operation gives an invariant pattern. Another signature of this invariance is the power-law distribution of many quantities, all related to the correlation length.

While this is a powerful technique for getting the relevant parameters at the transition, the chance of observing power-laws originating from a phase transition in nature seems to be quite low, due to the necessity of carefully adjusting control parameters. However, in many cases (earthquakes, waves, brain dynamics, and social organizations), fluctuations with a power-law distribution can be observed [78]. As illustrated by the Bak–Tang–Wiesenfeld model [79], a slowly driven system may behave in this way, giving origin to the expression "Self-Organized Criticality" (SOC) [80]. A similar behavior is also sown by systems whose evolution is given by some extremal value, such as the maxima of some quantity over all the lattice [81]. Finally, one can try to control the location of the critical point by replacing the random processes with correlated ones, as in explosive percolation [82,83].

Actually, a method for mapping a percolation problem into a SOC one had been discovered before, and is named invasion percolation [84]. First, extract all needed random numbers, attaching them to the sites of the space-time lattice as a quenched random field. Then, suppose to slowly increase the percolation probability, marking all sites (or links, according with the problem) with a random number greater than $p$. If the percolation probability is a monotonous function of the number of "wet" neighbors, the clusters will keep growing with increasing values of $p$, until a percolative cluster appears. If one now starts from a given set of wet sites, for instance one boundary of the system, and looks for the minimum value of the random numbers linked to connecting sites, this is the value of $p$ for which this site will be added to the existing cluster. After adding this site, one should also add all

connected sites with random number less than the established value of $p$. When no more sites can be added, a new search for the minimum on the boundary is performed (this minimum is necessarily larger than $p$), and the process is repeated until the opposite side is reached.

This procedure gives the minimum number of connected sites for percolation, what is called the "backbone" of the percolation cluster. To illustrate the basic idea with a working procedure in case of processing of information, let us consider a simple bond percolation problem, with a given fixed infection probability [85]. There is a set of $N$ nodes, $x_i$, that can stay in two states: 0 for "healthy" and 1 for "tainted" (or contaminated). Node $i$ processes information coming from another node $j$, defined by an adjacency matrix $a_{ij} = 1(0)$ for connected (disconnected) nodes. We assume that node $i$ is tainted and it can "infect" other nodes with a probability $\tau$.

The epidemic process is simulated by computing, for each node $i$ and time $t$, the state $x_i(t)$ by taking the OR ($\vee$) of the infection process along each connection, where the single infection event from node $j$ to node $i$ is computed by extracting a random number $r_{ij}$, evenly distributed between 0 and 1, and comparing it with $\tau$, i.e.,

$$x_i(t+1) = \bigvee_{j=j_1^{(i)},\dots,j_{k_i}^{(i)}} [\tau > r_{ij}(t)] x_j(t), \tag{10}$$

where $\bigvee$ represents the OR operator and the multiplication represents the AND. The square bracket represents the truth function, $[\cdot] = 1$ if "$\cdot$" is true, and zero otherwise. The quantity $r_{ij}(t)$ is a random number between 0 and 1, drawn independently for each triplet $i, j, t$.

To obtain an equation for the epidemic threshold, let us replace $x_i(t)$ by $[\tau > \tau_i(t)]$ (or $[\pi > \pi_i]$ for the site problem). The quantity $\tau_i(t)$ represents the minimum value of the infection probability $\tau$ for which site $i$ is infected at time $t$. Equation (10) becomes

$$[\tau > \tau_i(t+1)] = \bigvee_{j=j_1^{(i)},\dots,j_{k_i}^{(i)}} [\tau > r_{ij}(t)][\tau > \tau_j(t)]. \tag{11}$$

Now, $[\tau > a][\tau > b]$ is equal to $[\tau > \max(a,b)]$ and $[\tau > a] \vee [\tau > b]$ is equal to $[\tau > \min(a,b)]$. Equation (11) can therefore be expressed as

$$[\tau > \tau_i(t+1)] = \left[\tau > \left(\mathop{\mathrm{MIN}}_{j=j_1^{(i)},\dots,j_{k_i}^{(i)}} \max\left(r_{ij}(t), \tau_j(t)\right)\right)\right],$$

and we get the desired equation for the $\tau_i$'s

$$\tau_i(t+1) = \mathop{\mathrm{MIN}}_{j=j_1^{(i)},\dots,j_{k_i}^{(i)}} \max\left(r_{ij}(t), \tau_j(t)\right). \tag{12}$$

Let assume that at time $t = 0$ all sites are infected, such that $x_i(0) = 1\ \forall \tau$, thus we write $\tau_i(0) = 0$. We can iterate Equation (12) and get the asymptotic distribution of $\tau_i$. The minimum of this distribution gives the critical value $\tau_c$ for which there is at least one percolating cluster with at least one "infected" site at large times, i.e., there is an epidemic spreading in the whole system. A schematic representation of this modus operandi is illustrated in Figure 14 (left).

Alternatively, one can study the site percolation process, where node $x_i$ first processes all incoming information and then probabilistically ($\pi$) checks the result. In this case, the dynamics is

$$x_i(t+1) = [\pi > r_i(t)] \bigvee_{j=j_1^{(i)},\dots,j_{k_i}^{(i)}} x_j(t), \tag{13}$$

and the evolution equation for the probabilities $\pi_i$ is

$$\pi_i(t+1) = \max\left(r_i(t), \underset{j=j_1^{(i)},...,j_{k_i}^{(i)}}{\text{MIN}} \pi_j(t)\right). \tag{14}$$

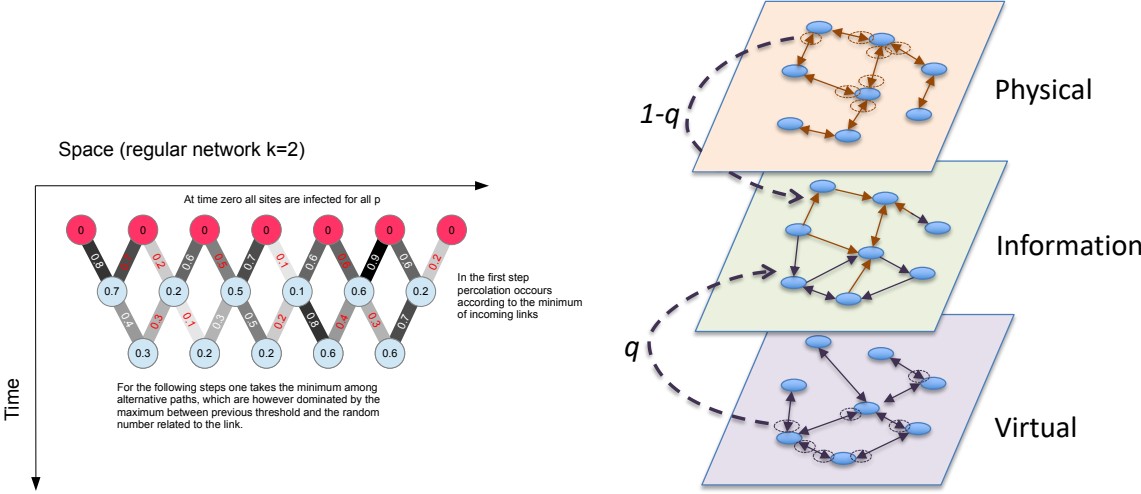

**Figure 14.** (**Left**) Evolution of the local minimum value of the percolation parameter $p_i$ for a 1D regular network with $k = 2$; and (**Right**) an example of multiplex generated with our method. We start with the physical and virtual networks, both symmetric and with the same average connectivity, and then we build the information network by choosing, for each node, outgoing links with probability $q$ from the virtual network and with probability $1 - q$ from the real one. Figures from [86].

## 6. Risk Perception

This method can be extended to cases in which a node has a knowledge of the local infection level, and can therefore lower the infection threshold according with the perception of the risk [85,87,88].

For modeling this case, we assume that the infection probability $\tau$ is replaced by a probability $u(s, k_i)$ that a site $i$ with connectivity $k_i$ is infected by any one of its $s$ infected neighbors as

$$u(s, k_i) = \tau f(s; J),$$

where $\tau$ is the "bare" infection probability and $f(s; J)$ is a monotonic decreasing function of the number of infected neighbors $s$, depending on some parameter $J$. For instance, in [89], the probability $u(s, k_i)$ is assumed to be

$$u(s, k_i) = \tau \exp\left(-J\frac{s}{k_i}\right),$$

The idea is that the perception of the risk, given by the percentage of infected neighbors and modulated by the factor $J$, effectively lowers the infection probability because the node checks the received information against the central server, paying the delay.

Due to the monotonicity of $f$, it is possible to invert the equation and obtain the explicit level of the epidemic threshold,

$$J_i(t+1) = \underset{j=j_1^{(i)},...,j_{k_i}^{(i)}}{\text{MAX}} \min\left(-\frac{k_i}{s_i(J_j(t))} \ln\left(\frac{r_{ij}(t)}{\tau}\right), J_j(t)\right),$$

where, analogously to the previous case, the critical value of $J_c$ is obtained by taking the maximum value of the $J_i(t)$ for some large (but finite) value of $t$.

Finally, it is also possible to consider the case in which the information comes from a network which can be partially different from that where the epidemic spreads.

This system is well represented as a multiplex network [90–100], i.e., a graph composed by several layers in which the same set of $N$ nodes can be connected to each other by means of links belonging to different layers.

Referring to Figure 14 (right), the information layer represents the perception network where people become aware of the epidemic thanks to news coming from virtual and physical contacts in various proportions. The physical layer represents the information network where the epidemic spreading takes place. The information layer is made by combining with probability $q$ links from the physical and virtual ones. The quantity $q$ therefore is a measure of the differences between the physical and information layers. In this case, it is also possible to obtain the epidemic threshold using a closed equation [86]. Clearly, if the networks are too different, the epidemics cannot be stopped for any precaution level $J$, as shown in Figure 15 (left).

As we can see in Figure 15 (right), for a physical random and virtual scale-free network, this transition between the values of $q$ for which the infection can be arrested by an increased precaution level and it is unstoppable is quite sharp, especially for low values of the bare infectivity probability $\tau$. A similar scenario holds for a mixture of physical and virtual random networks [86]. This results has to be considered in a practical implementation, since the most convenient level of precaution is that near the epidemic threshold, but when the information is not representative of the real risk (large values of $q$), the epidemics may become unstoppable.

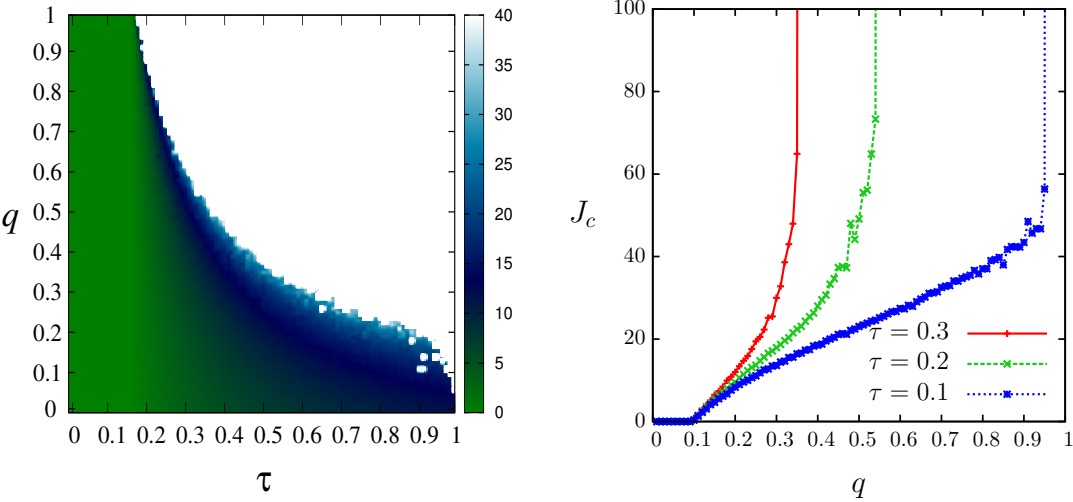

**Figure 15.** (**Left**) Critical precaution threshold $J_c$ (gray/color code) as a function of the bare infection $\tau$ and of the difference $q$ between the physical and the information network. Here, the physical and virtual networks are Poissonian (random) with $\langle k \rangle = 6$ and $N = 1000$. In the darker region, there is always a value of $J_c$ able to stop the epidemics, while in the white region the epidemics cannot be stopped. The separation boundary is the "stoppability" frontier. (**Right**) Critical precaution threshold $J_c$ versus the difference between the physical and the information network $q$ for some values of the bare infection $\tau$ (from right to left $\tau = 0.1, 0.2, 0.3$). Random physical network and scale-free virtual network, both with $\langle k \rangle = 6$, $N = 10,000$. Figures from [86].

In summary, our method allows to obtain the epidemic threshold in just one run, without having to repeat the simulation with many tentative infection probabilities, looking for the outbreak threshold. It can be considered an example of self-organized criticality [80], in which a system automatically discovers the critical value of a parameter; in particular it is very reminiscent of the Bak–Sneppen evolutionary model [81]. It can be directly implemented in computer networks, allowing nodes to exchange also their estimated epidemic threshold.

Epidemic/percolation models are characterized by a monotone increasing of the probability of being contaminated with the number of infected neighbors, and this characteristic allows to explicitly obtain the epidemic threshold by our self-organized critical method. We can also consider other processes, for instance with an "interference" among infective agents, and in general processed based on generic local rules like cellular automata [43]. However, in these cases, the monotonicity is lost and one has to consider more complex data structures [101].

This difference can be visually inspected comparing the two plots of Figure 16. The black regions in the figures shows the values of the probability $\pi$ for which the epidemic process cannot reach site *i*. The value $\pi_i$ of Equation (14) corresponds to the lowest end of the bar, and the the critical value for the epidemic probability is the smallest value of $\pi_i$.

In Figure 16 (right), the equivalent regions for a "nonlinear" process in which the OR of Equation (13) is replaced with a XOR is shown. The regions are no more compact, and therefore one cannot simply represent them with a single number. Multibit storage can be used in this case [101].

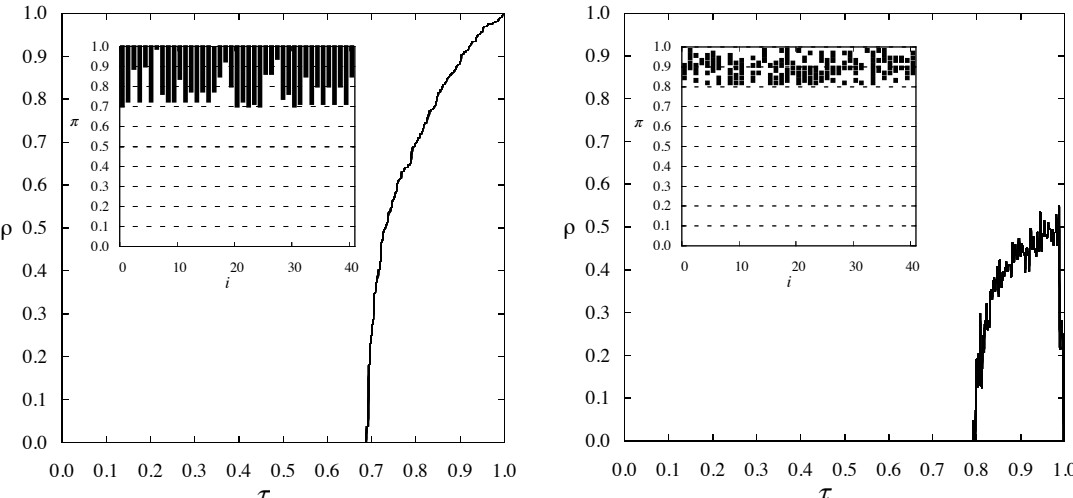

**Figure 16.** Average density ($\rho$) as a function of the bare infection probability $\tau$ and in the inset the distribution of the values of $\pi_i$ for which the infection reaches site *i* (black bar) for two percolation problems in a regular network with $k = 2$: (**Left**) site percolation; and (**Right**) "nonlinear" (XOR) percolation. Figures from [101].

## 7. Chaos, Synchronization and Control

The final percolation problem that we want to address in this review is related again to the sensitivity with respect to the variation in the initial configuration. We show in Section 3 that this is not always the case: in some regions (absorbing and disordered), any initial variation is reabsorbed, while, in the chaotic region, statistically even a small perturbation extends to the whole system (damage spreading) [69,102]. It is therefore another example of percolation, and indeed the two phenomena belongs to the same universality class [103].

The propagation of an initial defect is analogous to the sensitive dependence on the variation of initial conditions typical of chaotic systems, and indeed it is possible to define the equivalent of Lyapunov exponents also for discrete systems [104,105].

Another approach to the characterization of the chaotic property or a discrete system is that of quantifying the effort needed to synchronize two copies of the same system, over the same quenched field (if present). In other words, two replicas of a chaotic system tend to diverge, thus, by measuring how strongly one should "push" the two system to synchronize them, one can get an estimation of the divergence. Indeed, the "chaotic" properties of cellular automata are related to the synchronization thresholds [105].

An interesting application of these considerations is that of exploiting the "chaotic" properties of a dynamical system to drive it towards a desired state. Since a chaotic system is extremely sensitive to perturbations, it can be driven with small efforts, given the possibility of computing, in real time, the directions of the divergence with respect to the desired ones [106]. A tentative applications of such concepts to cellular automata and other percolation problems can be found in [107–112].

## 8. Conclusions

In this review, we discuss some aspects of percolation problems that can be useful for Internet and computer scientists. We introduce a common language for both static and dynamic percolation processes, showing their relationship with infection and opinion dynamics models. We then illustrate their relationship with the Monte Carlo implementation of equilibrium models, such as the Ising and the Potts ones, which have often been exploited as a background for opinion influences. From there, we expand the problem to include non-equilibrium processes, in particular cellular automata (CA), in both the deterministic and stochastic versions, showing that the probabilistic CA can be seen as deterministic ones over a quenched random field. In particular, we examine the role of absorbing states. After reviewing some properties of opinion models with norms, and specifically the presence of absorbing, disordered and "chaotic" phases, we compare the actual evolution on a lattice with the mean-field results.

After that, we consider the influence of the network structure, showing that the rewiring of local links allows for a smooth transition from regular lattices to random graphs. We exploit this transition showing that these change in the topology can trigger a cascade of bifurcations when a mean-field approximation corresponds to a chaotic map.

We then consider the problem of determining the percolation thresholds using a distributed algorithm. By comparison with the self-organized criticality and invasion percolation, we illustrate a method for determining the threshold for many problems, including those where the perception of the risk of being infected affects the infection probability itself, and considering also the case of different networks for the diffusion of epidemics and of the related information.

Finally, we illustrate the widely open problem of controlling percolation and other processes in distributed systems.

**Funding:** This research received no external funding.

**Acknowledgments:** RR acknowledges partial support from INFN.

**Conflicts of Interest:** The authors declare no conflict of interest.

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
