# Peer review of "Percolation and Internet Science"

_futureinternet, doi:10.3390/fi11020035_

Round 1

Reviewer 1 Report

In this work, the authors present a model of opinion spreading, an analyze the role of the topology of the network in inducinh coherent oscillations and the influence of risk perception for stopping epidemics. They also introduce the open problem of controlling percolation and other processes on distributed systems.

The paper is well-written an organized, and the content of the paper sounds interesting. I suggest the authors an accurate check of the English style for eliminating some errors an typos and improving in some place the Eglish form that is, in overall, acceptable.

Author Response

We have revised the english correcting many mistakes and typographic errors. 

Reviewer 2 Report

The manuscript by Bagnoli, Bellini, Massaro, Rechtman, presents a really interesting review encompassing an exhaustive investigation of the percolation problem and the aspects referred to it, and its generalization cellular automata. The authors choose to explore some of the multidisciplinary fields, linked to percolation definition, able to shed light on how these topics can shape the basis of the Internet Science. 

The structure of the manuscript is clear and all the sections presented are based on a wide scientific literature. 

The manuscript is well-written, clear and concise, and English is comprehensible and satisfactory 

The figures are relevant with adequate captions.

I think that this work can be considered for publication in Future Internet but I have a very minor comment:

Referred to the following sentence in the abstract “Originated in the domain of theoretical and matter physics, it has many applications in epidemiology, sociology and, of course, computer and Internet sciences.” 

I expect to find in the argued sections, from one hand the focus mainly on the physics point of view but from the other hand I thought the authors would discuss also about technological aspects of Internet Science, beckoning Information and Communication Technology applications, or healthcare procedures, cooperation dilemma, evolutionary game theory, social contagion. I would advise authors to add some references about examples of application domains as the final target of Internet Science. 

Author Response

We have revised the English, correcting errors and mistakes. We also added a new paragraph about technological aspects of Internet Sciences related to the percolation issue, and rearranged some material for a smoother reading, avoiding some repetitions.